# REWRITING PROTEIN ALPHABETS WITH LANGUAGE MODELS

**Lorenzo Pantolini**[1,2]**, Gabriel Studer**[1,2]**, Laura Engist**[3]**, Ieva Pudžiuvelytė**[1,2]
**Florian Pommerening**[3]**, Andrew Mark Waterhouse**[1,2]**, Stefan Bienert**[1,2]**, Gerardo Tauriello**[1,2]
**Martin Steinegger**[4]**, Torsten Schwede**[1,2]**, Janani Durairaj**[1,2]

[1]Biozentrum, University of Basel, Basel, Switzerland
[2]SIB Swiss Institute of Bioinformatics, Basel, Switzerland
[3]Department of Mathematics and Computer Science, University of Basel, Basel, Switzerland
[4]School of Biological Sciences, Seoul National University, Seoul, South Korea
`{firstname.lastname}@unibas.ch`     `martin.steinegger@snu.ac.kr`

## ABSTRACT

Detecting remote homology with speed and sensitivity is crucial for tasks like function annotation and structure prediction. We introduce a novel approach using contrastive learning to convert protein language model embeddings into a new 20-letter alphabet, TEA, enabling highly efficient large-scale protein homology searches. Searching with our alphabet performs on par with and complements structure-based methods without requiring any structural information, and with the speed of sequence search. Ultimately, we bring the exciting advances in protein language model representation learning to the plethora of sequence bioinformatics algorithms developed over the past century, offering a powerful new tool for biological discovery.

## 1 INTRODUCTION

Protein sequence alignment has been a cornerstone of bioinformatics for decades, with its use growing significantly alongside expanding databases and deep learning advances in structure and function prediction (Steinegger & Söding, 2017; Kallenborn et al., 2025; Jumper et al., 2021; Mirdita et al., 2022). Methods like BLAST (Altschul et al., 1990) (Basic Local Alignment Search Tool) revolutionised database searching by using heuristics to find regions of local similarity, offering a fast, albeit less sensitive, alternative to exhaustive dynamic programming algorithms. Tools like MMseqs2 (Steinegger & Söding, 2017) (Many-against-Many sequence searching) led to further massive speed improvements by introducing highly efficient indexing techniques and $k$-mer matching strategies to accelerate the initial search phase, making it possible to compare sequences against massive protein databases in minutes. The increasing availability of GPUs has also played a role in optimisations, with MMseqs2-GPU (Kallenborn et al., 2025) now achieving high speeds without the need for $k$-mer matching. However, sensitivity remains a critical objective in sequence comparison, serving two essential roles. High sensitivity is necessary not only for finding distant evolutionary relationships when close homologs are scarce, but also for aiding protein modelling efforts based on Multiple Sequence Alignments (MSAs). As demonstrated across numerous studies, more sensitive profile-based MSAs (e.g using JackHMMER (Finn et al., 2011; Johnson et al., 2010)), and in general increasing the depth of an MSA can lead to the construction of significantly better predicted structures (Odai et al., 2025; Zheng et al., 2024; Kim et al., 2025). Naturally, these more sensitive search strategies incur a higher cost in computational time, presenting a persistent trade-off between speed and depth of search.

Simultaneously, the recent years of access to unprecedented amounts of high quality predicted structures by methods such as AlphaFold2 (Jumper et al., 2021) has accelerated innovation. The fact that protein structure is more conserved than sequence has driven efforts to incorporate structural information into sequence comparison frameworks. For example, Foldseek uses a structure-based alphabet called 3Di (Three-Dimensional Interaction) in a fast and sensitive sequence alignment

approach to detect remote homology between sequences with similar folds (Van Kempen et al., 2024). More recent works such as ESM3 (Hayes et al., 2025) and FoldToken (Gao et al., 2025) have also used structural tokenization, but these typically create thousands to hundreds of thousands of tokens with the aim of faithful structure reconstruction, too sparse to use in sequence comparison frameworks. However, most sequence databases lack corresponding high-confidence structural models. For instance, billions of protein sequences are available in databases like UniProt (Consortium, 2019), MGnify (Richardson et al., 2023) and now LOGAN (Chikhi et al., 2024). In contrast, far fewer experimentally determined protein structures exist in the PDB (Burley et al., 2017). While predicted structure databases like the AlphaFold Database (AFDB) (Varadi et al., 2024) and the ESM Atlas (Lin et al., 2023) help close this gap, a large fraction of their residues are structurally repetitive or have low confidence. In addition, storing and searching databases of millions of predicted structures requires significant resources and compute. This disparity limits the applicability of structure-based methods across the full breadth of sequence space. Furthermore, structural comparison is suboptimal for rapidly evolving proteins with poor predictions and for proteins with extensive disordered regions that complicate structural alignment.

In parallel, protein language models (pLMs) have emerged (Elnaggar et al., 2021; Brandes et al., 2022; Lin et al., 2023), excelling at remote homology detection by generating high-dimensional embeddings that capture complex evolutionary and structural information. The power of these models lies in representation learning: their ability to compress the vast sequence space into dense, numerical vectors (embeddings) where evolutionary and functional relationships are represented. Recently, pLM embeddings have been directly harnessed for remote homology detection. This includes using sequence-level aggregated representations for fast comparison, such as in ProtTucker (Heinzinger et al., 2022) and TM-vec (Hamamsy et al., 2024), where the distance between two sequence embeddings is used as a proxy for evolutionary distance. Other methods, such as EBA (Pantolini et al., 2024) (Embedding-Based Alignment) and pLM-BLAST (Kaminski et al., 2023) leverage the per-residue embeddings to generate sequence alignments. While aligning these continuous representations at the residue-level is highly sensitive, it is no match to the speed and scalability of discrete sequence alignment.

Here, we trained a shallow neural network to discretise pLM embeddings, generating a new 20-letter alphabet, TEA (The Embedded Alphabet), to express protein sequences. Our new alphabet retains the exceptional remote homology detection of pLMs while enabling large-scale comparisons using highly optimised alignment tools like MMseqs2. It performs on par with structure-based approaches and complements them in cases where structural data is suboptimal. Furthermore, TEA provides a confidence metric, offering a valuable estimate of prediction reliability for downstream analysis.

## 2 RESULTS

### 2.1 REWRITING THE LANGUAGE OF LIFE WITH CONTRASTIVE LEARNING

Embedding discretisation can be achieved in various ways. A straightforward approach involves predicting a probability vector, sized to the desired alphabet, and then converting it to the character with maximum probability. This method naturally aligns with the typical training objective of protein language models (pLMs), which often reconstruct protein sequences from embeddings by predicting token probability distributions (including amino acids). However, by modifying the training objective, we can learn a new representation that forces alignment of residues onto the same characters, thereby enabling the comparison of remote homologs.

To achieve this, we trained a shallow discretisation head (depicted in Figure 1A) to convert pLM embeddings into logit representations, and subsequently character probabilities, which are then transformed into discrete characters. This head was trained using a contrastive learning objective designed to force similar character probabilities for structurally aligning residues from homologs, while pushing dissimilar probability vectors for non-aligning residues. As described in Figure 1A, we created residue triplets from structurally aligning residues: an anchor, a positive (the anchor's structurally aligning counterpart), and a negative (a randomly selected residue from a 5-residue window around the aligning position). We also incorporated losses favouring a uniform character distribution, and low Shannon entropy across the predicted probabilities. The training paradigm is described fully in Section A.2. This training forces the network to produce characters that result in higher sequence identity for residues sharing similar structural characteristics, producing an alphabet

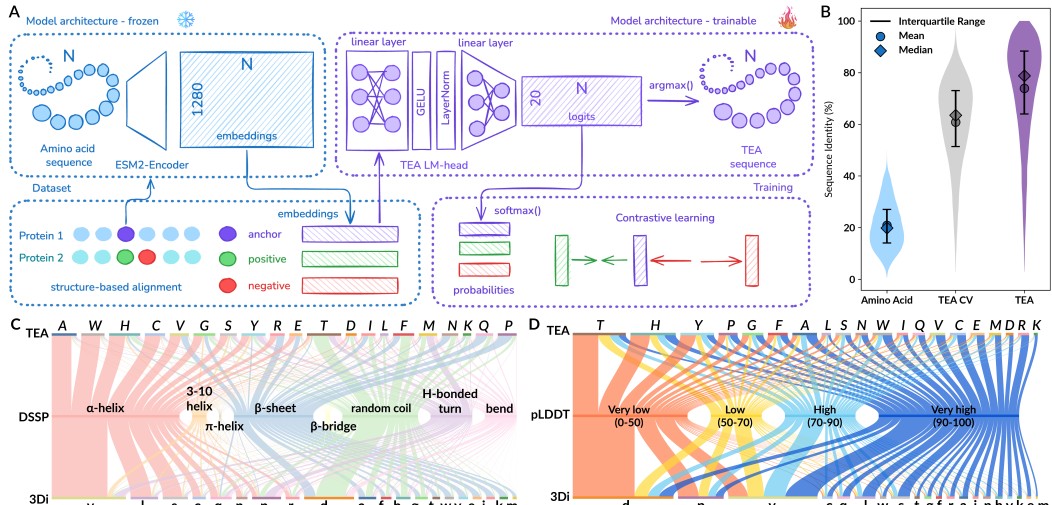

Figure 1: **Model and training. A)** Overview of the model architecture and training regimen. **B)** Sequence identities derived from 24,814 structural alignments (max. 100 per superfamily) of proteins in the same SCOPe family having sequence identity <40%, for amino acids and TEA converted sequences. TEA CV represents a 4-fold cross-validation across SCOPe40 folds described in Section A.1.1, where sequence identities are reported for the alignments from the held-out fold. **C)** A Sankey diagram depicting the distributions and comparisons of (1) TEA characters, (2) DSSP secondary structure labels, and 3) 3Di characters for 15,128 proteins from SCOPe40 and 10,000 proteins from AFDB with average pLDDT>90. **D)** A Sankey diagram depicting the distributions and comparisons of (1) TEA characters, (2) AFDB pLDDT, and 3) 3Di characters for 40,000 proteins from AFDB with 10,000 from each pLDDT bin.

that enables comparison of remote homologs. As shown in Figure 1B, while the pairwise amino acid sequence identity of SCOPe40 alignments is low, the TEA sequence identity is much higher. This holds true even for entire protein folds not included in the training set, demonstrating the generalisability of our alphabet to sequences and folds not seen by our model.

Given the similarity in approach of discretising continuous data into a 20-letter alphabet, we sought to ascertain whether we learn similar patterns as the 20 characters representing the 3Di alphabet underlying Foldseek, built by vector quantization of protein structural information (Van Kempen et al., 2024). TEA characters correlate strongly with secondary structure annotations (Figure 1C rows 1 and 2)[1]. 3Di, like TEA, also has heavy association with secondary structural elements (Figure 1C rows 2 and 3) but differs in terms of distribution and numbers of characters assigned to the different elements. In fact, a majority of residues in helices are assigned to the 3Di-v and loops to 3Di-d, which are known to be overrepresented in larger datasets (Heinzinger et al., 2024), while multiple letters in TEA all represent helices (*A, W, H, C, G* etc.), sheets (*P, Q, K,* etc.) and loops (*T, F, M* etc.). As shown in Figure 1D (rows 2 and 3), 3Di by definition relies on the quality of the underlying structural data used for construction with most low pLDDT residues manifesting as disordered loops and thus mapping to 3Di-d (54%). Meanwhile, TEA characters are more uniformly spread Figure 1D (rows 1 and 2), with only 29% of low pLDDT residues mapping to *T, F,* and *M*. Some examples of common fully conserved TEA motifs of length 17 are illustrated in Supplementary Figure 5. These motifs are well-conserved in secondary structural elements despite low conservation at the amino acid level, and the motifs themselves range from being found across different protein folds to only being found within a certain protein family. This highlights a promising opportunity for motif discovery to detect structural or functional similarities.

---

[1] TEA characters are shown in italics and 3Di in lowercase; the amino acid letters are used for depiction purposes but no direct amino acid correspondence is expected for either alphabet.

## 2.2 TEA SEARCHES REACH STRUCTURAL ACCURACY AT A FRACTION OF THE COST

We benchmarked our alphabet's remote homology and structural similarity detection performance using the SCOPe40 and multi-domain benchmarks from the Foldseek paper (Van Kempen et al., 2024). We used MMseqs2 with sequences converted to our TEA alphabet and a custom substitution matrix (see Section A.3). As shown in Supplementary Figure 7, and in contrast to the BLOSUM matrix, many off-diagonal elements of the TEA substitution matrix are highly negative, indicating orthogonal characters. As each of the three alphabets, amino acid (AA), 3Di, and TEA come with their own associated substitution matrix, the substitution scores can be combined during the alignment stage to potentially boost performance (see Section A.4 for implementation details).

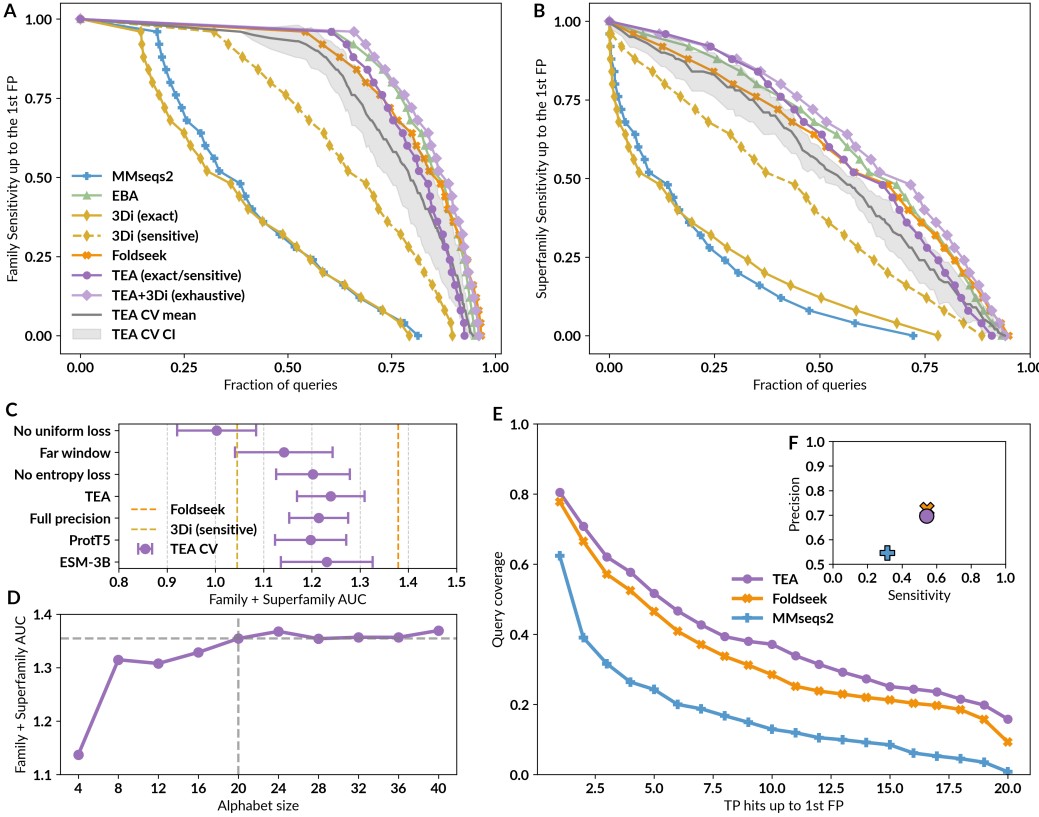

Figure 2: **TEA reaches similar sensitivities as structural aligners. A)** Cumulative distributions of sensitivity for homology detection on the SCOPe40 database of single-domain structures. TPs are matches within the same family; FPs are matches between different folds. Sensitivity is the area under the ROC curve up to the first FP. **B)** Same as A with TPs as matches within the same superfamily. **C)** Sum of cross-validation AUCs of family and superfamily curves (mean and confidence intervals) for various model ablations (see Section A.2.1), with the AUC for 3Di (sensitive) and Foldseek shown as yellow and orange dotted lines respectively. CV results in A-C are shown only for held-out SCOPe folds. **D)** Sum of family and superfamily curve AUCs shown for models trained with different alphabet sizes. The final 20-character TEA model is indicated with dotted lines. **E)** Search sensitivity of one hundred queries against 56,574 multi-domain, full-length AlphaFold2 protein models. Per-residue query coverage (y axis) is the fraction of residues covered by at least x (x axis) TP matches (LDDT > 0.6) ranked before the first FP match (LDDT < 0.25). **F)** Alignment quality for alignments from E, averaged over the top five matches of each of the 100 queries. Sensitivity = TP residues in alignment/query length; precision = TP residues/alignment length.

As Figure 2A-B shows, TEA (purple circles) achieves sensitivity comparable to Foldseek (Van Kempen et al., 2024) (orange crosses) and raw embedding-based alignment (Pantolini et al., 2024) (green triangles) for detecting relationships at both family and superfamily level. MMseqs2 and Foldseek employ a sophisticated similar $k$-mer generation technique by default, enabling sensitive searches at

the expense of speed. Using exact $k$-mer matching instead causes a drastic decrease in performance when running MMseqs2 with the 3Di alphabet and substitution matrix (yellow diamonds, solid vs. dashed lines). Conversely, TEA yields identical performance in both exact and sensitive modes (purple circles), and reaches structure-level sensitivity with exact $k$-mer matching, underscoring the intrinsic representation power of the alphabet. Supplementary Figure 6 shows the performance of an exhaustive all vs. all alignment without any pre-filtering step for different alphabets and combinations of alphabets. The performances of 3Di (yellow diamonds), 3Di+AA (light yellow pentagons) and Foldseek (orange crosses) clearly demonstrate that amino acid integration and further ranking optimisations are all crucial for Foldseek performance. Foldseek ranking optimisations include, for example, compositional bias correction and reverse bit score addition to suppress FP alignments, and multiplying the bit score with alignment LDDT and TM-score to obtain a structural bit score for ranking (Van Kempen et al., 2024). While the latter requires structures, the former FP-suppression measures could similarly further boost TEA performance.

Overall, TEA achieves Foldseek-level performance with exact $k$-mer matching mode, and without any alphabet combinations or ranking optimisations, thus delivering results with faster algorithms than typical amino acid searches at structure-level sensitivity (see Supplementary Table 2). Furthermore, the results from a 4-fold SCOPe fold-level cross-validation (in gray) confirm that TEA remains competitive with structure-based searches even on unseen protein folds. Excitingly, the combination of 3Di with TEA (light purple diamonds in Figure 2A) performs better than either individually, pointing to complementarity between the two representations. We leave the development of search tools making use of such combinations and further ranking optimisations for future work. In all search results reported from this point, unless otherwise specified, TEA is run with exact $k$-mer matching mode. We justify our model and training choices through ablations run on the 4-fold SCOPe fold CV, with results depicted in Figure 2C and described in detail in Section A.2.1.

Moving beyond single domain proteins, Figure 2E-F presents search sensitivity and alignment quality results for full-length multi-domain AFDB proteins. To avoid predicted disorder, we restricted the dataset to proteins with an average pLDDT > 90. All assessments were reference-free, based solely on the LDDT of the resulting alignments. Across both benchmarks, TEA demonstrates accuracy similar to Foldseek, and much higher than MMseqs2 using the standard amino acid alphabet.

To enable using TEA as a search service with meaningful E-values, we developed STEAM (Search with TEA against Many) which combines TEA and amino acid substitution scores during alignment and outputs E-values using a log-linear model as described in Edgar & Sahakyan (2025) (see Section A.6 for details). We developed a webserver (`https://pickybinders.org/tea`) for multi-database searches, further described in Section A.7.

## 2.3 ENTROPY AS A MEASURE OF CONFIDENCE AND STRUCTURE PREDICTION

Associating predictions with reliable measures of confidence is very important, especially when using methods that are difficult to interpret, such as neural networks. A successful example is predicted LDDT (pLDDT), a metric provided by modern protein structure prediction methods like AlphaFold2/3 (Jumper et al., 2021; Abramson et al., 2024) and ESMFold (Lin et al., 2023). The pLDDT score estimates the local distance difference test (LDDT) (Mariani et al., 2013) of a predicted model against a hypothetical ground truth, providing a per-residue confidence score. Previous research has shown that pLDDT can correlate both with the inability of the neural network to model the structure or with the presence of intrinsically disordered regions (Piovesan et al., 2022).

While we did not explicitly train a confidence predictor, the normalised Shannon entropy (Equation 2 in Section A.2) associated with our model's probability vectors can be interpreted as a measure of its certainty about a specific character. We therefore explored the normalized Shannon entropy of a probability vector as uncertainty at residue level and the average of the per-position entropies as uncertainty at sequence level. We observe that average entropy correlates inversely with search sensitivity in the SCOPe40 benchmark (Figure 3A), establishing it as a useful confidence measure for search results. Specifically, sequences with average entropy below 0.25 ($n$=2,729) achieved an average sensitivity of $0.81 \pm 0.29$, whereas all sequences above this threshold ($n$=32) have zero sensitivity. In general, cases with low entropy (i.e. high confidence) but also low family sensitivity are typically difficult for both our method and Foldseek (Figure 3A bottom left, white), indicating that they represent challenging relationships for this benchmark.

We also observed a significant correlation between TEA entropy and ESMFold pLDDT. Figure 3B shows the entropy distribution for residues from different pLDDT bins. We note that while the correlation trend is clear, many low pLDDT residues have low entropy, indicating that predicted disordered regions within otherwise high-confidence proteins are still consistently represented by our alphabet. Interestingly, we observed a strong correlation between average entropy and average pLDDT with a Spearman correlation of -0.823 to the maximum of AlphaFold and ESMFold pLDDT for the proteins in the 40k-pLDDT set (see Section A.8). In Figure 3C-D, we show that entropy very often captures structural uncertainty, with high pLDDT proteins having low entropy (Figure 3C) and low pLDDT proteins having high entropy (Figure 3D). However, TEA is not limited to the accuracy of, for example, ESMFold which also uses pLM embeddings for structure prediction. We showcase two examples of such outliers with low TEA entropy in Supplementary Figure 8: the first case where the ESMFold pLDDT is low but AlphaFold pLDDT is high and second where the AlphaFold pLDDT is low and ESMFold pLDDT is high. In both cases, TEA search results against AFDB Clusters (Barrio-Hernandez et al., 2023) and the PDB (Burley et al., 2017) demonstrate that the TEA sequence represents the higher pLDDT structure. This suggests that our alphabet can confidently and accurately model structural features even when structure prediction methods are unconfident.

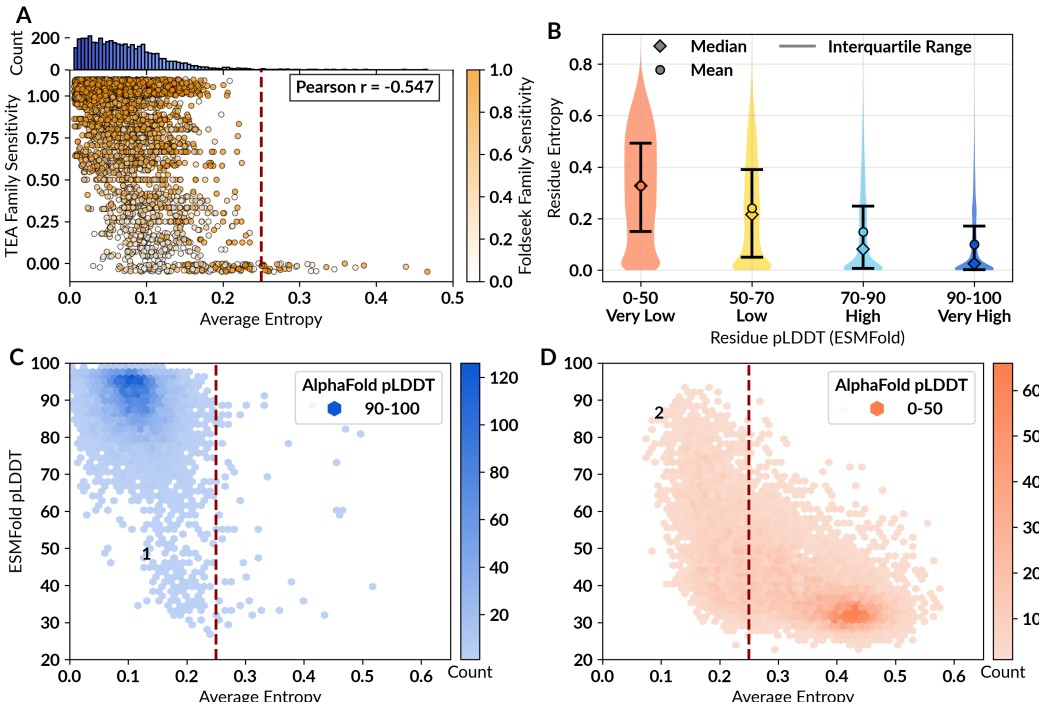

Figure 3: **Entropy as a measure of confidence. A)** Average TEA entropy vs. TEA family sensitivity up to the first FP from the SCOPe40 benchmark for all proteins from families of size 3 or more, coloured by the respective Foldseek family sensitivity. Points with family sensitivity of 0 and 1 have a small amount of jitter below 0 and above 1 respectively for visibility. **B)** Residue-level ESMFold pLDDT (binned) vs. residue-level TEA entropy for 40,000 residues from each pLDDT bin obtained from the 40k-pLDDT set (see Section A.8). **C)** Average TEA entropy vs. average ESMFold pLDDT for 10,000 proteins where the corresponding average AlphaFold pLDDT is over 90. **D)** Same as C but for 10,000 proteins where the average AlphaFold pLDDT is less than 50. Points labelled 1 and 2 are described further in Supplementary Figure 8.

## 2.4 IMPROVING FUNCTIONAL ANNOTATION WITH REMOTE CONNECTIONS BEYOND ALPHAFOLD2

Recently Foldseek was used to create the impactful AFDB Clusters dataset (Barrio-Hernandez et al., 2023), which groups over 200 million proteins from AFDB into 15.3 million non-fragment clusters. However, only 2.3 million of these clusters have more than one entry - the remaining 13 million are

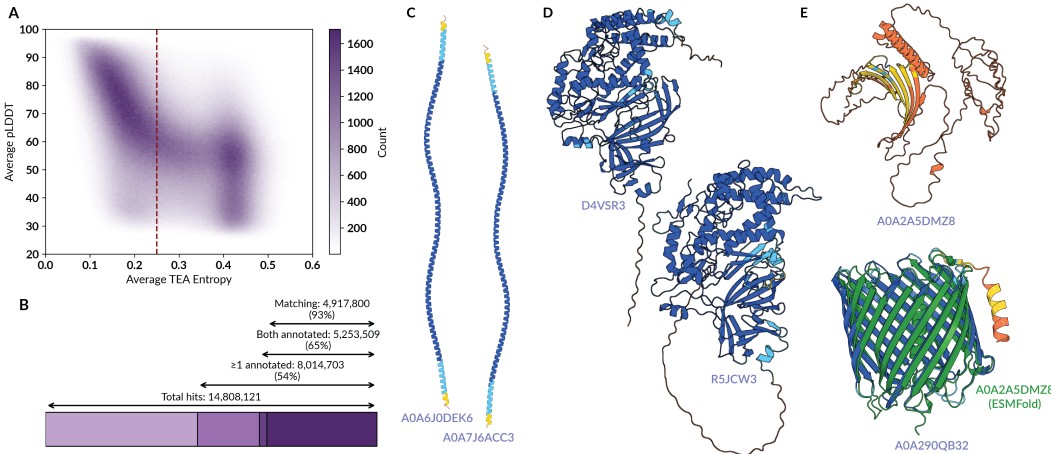

Figure 4: **Remote homology in structural singletons. A)** pLDDT vs. TEA entropy for all 15.3 million AFDB cluster representatives. The chosen entropy threshold is shown as a dashed red line. **B)** The total number of found hits at >50% TEA sequence identity and >90% coverage when searching singletons against all representatives (both with entropy <0.25), the number/percentage of hits where either query and target have annotations in InterPro (Blum et al., 2025), where both have annotations, and where at least one annotation matches between the two. Percentages reported are always with respect to the previous count. **C-E)** Examples of singletons and their closest found cluster representative, missed by Foldseek. **C)** Proteins with mainly helix content **D)** Proteins with disordered regions. **E)** Proteins with low-confidence AlphaFold2 models. Note that for A0A2A5DMZ8, the ESMFold (Lin et al., 2023) model (shown in green) is a $\beta$-barrel with <3Å RMSD to the AFDB model of the closest hit, supporting the structural similarity suggested by TEA.

singletons. We used TEA to connect these singletons to cluster representatives where possible. We used an average entropy threshold of <0.25 to select proteins where our alphabet would provide confident results (Figure 4A). This procedure filtered the dataset to 1.86 million representatives (81%) and only 5.24 million singletons (40%), indicating that many singletons are structurally and evolutionarily ambiguous both to Foldseek and pLMs, potentially pointing towards sequencing errors and protein fragments.

Searching with TEA with a 90% coverage threshold (as used in the original AFDB clustering) and 50% TEA sequence identity threshold successfully found over 14 million hits for over 1.5 million singletons. Of these new connections, 35% had InterPro annotations for both query and target, and 93% of those annotations matched exactly (Figure 4B), demonstrating the high accuracy of our hits. Excitingly, over 2.7 million hits have either query or target annotated, but not the other, pinpointing cases where functional annotation transfer could be performed. Figure 4C-E shows examples of newly connected singletons and their closest cluster representatives, all of which have exact InterPro matches. These examples highlight three common use cases where our method is effective: (1) proteins with mainly helix content, where Foldseek often struggles to find meaningful structural matches due to the over-representation of 3Di-v (see Figure 1C), but the underlying sequences and embeddings still show detectable structural homology (panel C), (2) proteins with disordered regions which cause shorter structural alignments failing the 90% coverage threshold (panel D), and (3) incorrectly modelled proteins where a structural search would fail (panel E). In the latter case, a comparable structure could be produced with ESMFold (Lin et al., 2023).

In summary, TEA generates highly relevant and complementary comparisons to structure-based methods while requiring minimal computational resources (<800 MB storage for >10 mil. sequences, 44 h for 35 mil. comparisons with 64 cores and 64 GB RAM). This proof of concept demonstrates that TEA similarities could be used with appropriate coverage and identity thresholds, perhaps even in combination with sequence and structure similarities, to recreate clustering efforts such as AFDB Clusters (Barrio-Hernandez et al., 2023) and UniProt3D (Durairaj et al., 2023) from a novel perspective, bringing new functional connections and discoveries to both domain-level and protein-level analyses.

## 3    DISCUSSION

We developed a new alphabet derived from protein language models that allows the identification of remote homologs with performance comparable to structure-based methods.

TEA offers several key advantages over existing protein sequence representations. As with standard amino acids, highly optimized and efficient sequence tools such as MMseqs2 can be applied, but, unlike amino acids, it facilitates the identification of remote structural homologs. Moreover, in contrast to structure descriptors like Foldseek's 3Di, our method requires no prior structural information, provides a built-in confidence measure via entropy, and ultimately complements structural representations by being more effective at comparing predicted disordered and repetitive regions. While we have shown that our alphabet works seamlessly with established sequence tools, further optimisations could improve ranking reliability. Additionally, it would be an exciting opportunity to test dedicated profile generation tools, such as HMMER (Finn et al., 2011), with TEA sequences. This would determine if the condensed and information-rich representation of our alphabet provides a significant boost to profile quality and coverage, or if this structural-evolutionary information is already fully captured by the alphabet's inherent capabilities. Such TEA-HMMs could also be useful for domain segmentation, providing a complementary alternative to approaches such as CATH (Sillitoe et al., 2021) and TED (Lau et al., 2024). Similarly, TEA sequences could provide useful and complementary insights for building and analysing phylogenetic trees, as 3Di has in FoldTree (Moi et al., 2025). An immediate and very useful application that we aim to explore is the creation of query-centric MSAs for AlphaFold modelling, where TEA searches could be transformative for modelling proteins currently showing shallow MSAs for traditional sequence search methods. Furthermore, further analysis of entropy could reveal regions of protein sequence space difficult to represent using pLMs and help improve future pLMs and representations.

Beyond TEA's immediate utility, the method provides a generalisable framework for training specialised alphabets from pLM embeddings using a contrastive objective. This architecture is readily adaptable, allowing researchers to fine-tune alphabets for specific goals. For instance, an alphabet could be trained for function prediction with characters grouping residues by their functional roles (e.g., active site, binding pocket) rather than purely structural characteristics. Another possibility is to develop an alphabet for interface description, enabling the clear characterisation of protein-protein interaction interfaces, which are often subtle and complex in continuous space. Alternatively, researchers could apply the same discretisation and contrastive learning principles to RNA sequences and their high dimensional descriptors (Wang et al., 2025), enabling highly efficient searches that might improve modelling efforts (Kretsch et al.).

Ultimately, TEA brings the powerful representation capabilities of deep learning to well-established sequence bioinformatics algorithms, such as profiles, phylogenetic trees, motif finding, multiple sequence alignments, and more, all while maintaining the speed and low resource consumption of amino acid sequences.

## 4    CODE AND DATA AVAILABILITY

The TEA model code, sequence conversion scripts, training scripts, and documentation are provided at github.com/PickyBinders/tea. TEA is also available on Hugging Face at huggingface.co/PickyBinders/tea. We provide TEA FASTA files of the following converted databases: AFDB Clusters, UniRef50 (version 2025_04), and PDB (version 25-11-12 SEQRES). A search server will be available at https://pickybinders.org/tea. Benchmarking data are available at https://doi.org/10.5281/zenodo.17725635.

## 5    ACKNOWLEDGEMENTS

We thank the members of the Schwede group for insightful discussions and technical support, and sciCORE at the University of Basel (https://scicore.unibas.ch/) for providing computational resources and storage space. We gratefully acknowledge financial support for parts of this work by the SIB Swiss Institute of Bioinformatics (https://www.sib.swiss/), the Biozentrum of the University of Basel (https://www.biozentrum.unibas.ch/), and the Swiss National Science Foundation (SNSF; Ambizione grant 223634 for JD, WEAVE grant 220141 for LP).

MEANINGFULNESS STATEMENT

We introduce a novel representation that explicitly captures the structural intuition hidden within protein language model embeddings. This work distils this latent knowledge into TEA, a discrete, 20-letter structure-aware yet structure-independent alphabet. We demonstrate that the essential structural and evolutionary signals of life can be compressed into a format that integrates seamlessly with decades of bioinformatics algorithms. This quite literally offers a new way to represent life (proteins), where remote homologs are expressed through similar sequences.

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

## A  Appendix

### A.1  Training data

Domains from SCOPe40 (v2.08) (Chandonia et al., 2022) were structurally aligned using TM-align and all alignments from within the same SCOPe40 superfamily with TM-score above 0.6 were retained. To generate triplets of residues, all aligning residue pairs with $C\alpha$ RMSD below 5 Å were considered as anchor and positive, with corresponding negatives for each anchor being randomly selected from a 5-residue window around the aligning position. The embeddings were obtained from the 4-bit quantized ESM2-650M model (`facebook/esm2_t33_650M_UR50D` on HuggingFace) (Rives et al., 2021).

#### A.1.1  Cross-validation

To assess generalisation potential, we also trained versions of TEA by four-fold cross-validation on SCOPe40, as described previously for Foldseek (Van Kempen et al., 2024). The SCOPe40 dataset is divided into four parts, such that all domains of each fold ended up in the same part of the four parts. The training data triplets were thus split into these folds and alphabets were trained on three parts and tested on the remaining part, selecting each of the four parts in turn as a test set. As good generalisation was seen in the cross-validation experiment, the final TEA model is trained on the entire training set.

### A.2  Model training

The embeddings from `facebook/esm2_t33_650M_UR50D` were discretized into an alphabet of 20 characters using contrastive learning on residue triplets, on an architecture consisting of a dense linear layer (1280x1280), a layer normalisation layer, a GELU activation function, and the final linear layer (1280x20) resulting in a model with 1.7 million trainable parameters. The model was trained for 10 epochs with the AdamW optimiser (Loshchilov & Hutter, 2017) (weight decay 0.3), a cosine annealing learning rate scheduler (from initial learning rate 0.005 to 0.0001), and a dropout of 0.1 added after the activation function.

The main, contrastive objective was to minimize the similarity between the anchor and the negative sample while maximizing the similarity between the anchor and the positive sample in each triplet. This approach inherently encourages the alphabet to represent increased similarity for aligning residues and decreased similarity for non-aligning ones. Given $\mathbf{a}$, $\mathbf{p}$ and $\mathbf{n}$ the probability vectors generated by our model for respectively the anchor, positive, and negative, the contrastive loss was implemented as follows:

$$\mathcal{L}_c(\mathbf{a}, \mathbf{p}, \mathbf{n}) = \frac{\mathbf{a} \cdot \mathbf{n}}{\|\mathbf{a}\| \, \|\mathbf{n}\|} - \frac{\mathbf{a} \cdot \mathbf{p}}{\|\mathbf{a}\| \, \|\mathbf{p}\|} \tag{1}$$

We also implemented two auxiliary losses: a Shannon entropy loss and a uniform loss. Lower entropy indicates a prediction strongly committed to a specific character, whereas higher entropy suggests the model is less certain about its character choice. We minimized the Shannon entropy associated to the probability vectors of the triplets with $N$ being the logits dimension (alphabet size):

$$H(\mathbf{x}) = -\frac{1}{N} \sum_i^N x_i \log(x_i), \quad \mathcal{L}_H(\mathbf{a}, \mathbf{p}, \mathbf{n}) = \frac{H(\mathbf{a}) + H(\mathbf{n}) + H(\mathbf{p})}{3} \tag{2}$$

To encourage uniform character usage, the uniform loss was achieved by applying a Kullback-Leibler (KL) divergence loss against a uniform distribution, $\mathbf{u}$. For each batch we averaged the probabilities relative to the predicted logits into a vector $\mathbf{b}$ and we computed a KL divergence loss as follows:

$$\mathcal{L}_u(\mathbf{b}) = \frac{1}{N} \sum_i^N b_i \log \left( \frac{b_i}{u_i} \right) \tag{3}$$

The final loss of the model was weighted as follows:

$$\mathcal{L} = \mathcal{L}_c + 0.5 \cdot \mathcal{L}_u + 0.1 \cdot \mathcal{L}_H \tag{4}$$

### A.2.1 ABLATIONS

To assess the impact of model and training choices, we trained a number of models with different settings ablated. These include *"No uniform loss"* and *"No entropy loss"* models where the $\mathcal{L}_u$ and $\mathcal{L}_H$ terms respectively in Equation 4 are removed, a *"Far window"* model where the negatives for each triplet were selected randomly from a window of 5-10 residues away from the aligning position instead of 1-5, a *"Full precision"* model where 4-bit quantisation is not used for embedding generation, and *"ESM-3B"* and *"ProtT5"* models where the pLMs used for embedding generation are changed to the `facebook/esm2_t36_3B_UR50D` model and the `Rostlab/prot_t5_xl_uniref50` model respectively, with corresponding changes in input embedding dimension. All of these ablations are trained on the 4-fold SCOPe fold split. Further, models with different alphabet sizes (4, 8, 12, 16, 24, 28, 32, and 40) were trained on the full training set.

Ablation results are depicted in Figure 2C. Enforcing uniform character usage through an explicit uniform loss (Equation 3) improves performance, as otherwise training favours fewer characters being utilised and thus reduces representation power (*No uniform loss* in Figure 2C). Crucially, moving the window further for selecting non-aligning (negative) residues for training worsens performance (*Far window*), as the embeddings of the obtained negatives become much easier to separate from the anchor and positive residues. Neighbouring residues, on the other hand, have very similar residue contexts and thus very similar embeddings to the anchor residue, forcing the model to learn a more nuanced separation between them, leading to better generalisation. Removing the explicit entropy loss term (Equation 2) also slightly worsens performance (*No entropy loss*). Using embeddings from a 4-bit quantized (Dettmers et al., 2021) ESM2 model did not show much difference compared to using full precision embeddings (*Full Precision*), thus unlocking fast and memory-efficient TEA sequence generation on consumer-grade GPUs even for longer proteins (0.05-0.5 seconds/sequence depending on length). We also find that full precision and 4-bit quantized embeddings can be used more or less interchangeably with TEA- using full-precision query TEA sequences against 4-bit quantized target TEA sequences in the SCOPe benchmark gave the same performance, and discrepancies in converted residues are seen only in residues with high entropy, thus also making TEA compatible with usage on CPUs. Using embeddings from the 3 billion parameter ESM2 model (*ESM-3B*) or using the ProtT5 language model (Elnaggar et al., 2021) (*ProtT5*) had overall similar results, demonstrating that the alphabetisation architecture and paradigm can in theory be applied to any current or future language model. Finally, as shown in Figure 2D, 20 is a reasonable choice for alphabet size, trading only minor performance increase for a more universally usable alphabet that can be straightforwardly plugged in to a number of protein sequence bioinformatics tools.

### A.3 TEA SUBSTITUTION MATRIX

We created a BLOSUM-like substitution matrix for TEA sequences from pairs of structurally aligned residues used for training. First, we determined the TEA states of all residues. Next, the substitution frequencies among TEA states were calculated by counting how often two TEA states were structurally aligned. (Note that the substitution frequencies from state A to state B and the opposite direction are equal.) Finally, the score for substituting state $x$ through state $y$ is $S(x, y) = 2 \log \frac{p(x,y)}{p(x)p(y)}$.

### A.4 ALIGNMENT METHODS

TEA is designed to be compatible with any classical alignment tool. In this work, we used MMseqs2 (version 18.8cc5c) (Steinegger & Söding, 2017) to perform alignments across three distinct modalities detailed below. When indicated as *sensitive*, we run the standard MMseqs2 easy-search command as follows:

```
mmseqs easy-search tea_query.fasta tea_target.fasta results.m8 tmp/
--comp-bias-corr 0 --mask 0 --gap-open 18 --gap-extend 3
--sub-mat matcha.out --seed-sub-mat matcha.out
```

For the *exact* modality, we include the flag `-exact-kmer-matching 1` in the easy-search command. This modification significantly increases the speed of the command by restricting the pre-filtering step to only allow exact $k$-mer matches. For 3Di and amino acids, gap open and gap

extend penalties of 12 and 1 were used, as these gave the best results on a grid-search for the SCOPe benchmark.

The sequences from models with different alphabet sizes (Figure 2D) and alphabet combinations (Figure 2A, Supplementary Figure 6) instead use all vs. all alignments with no pre-filtering step, followed by taking the top 2000 alignments per query (labelled *exhaustive*). These alignments were calculated using Needleman-Wunsch-style dynamic programming with gap penalties on score matrices, constructed such that $M_{ij}$ represents the subsitution score of residue positions $i$ and $j$. We used BLOSUM62 for amino acids, the 3Di substitution matrix for 3Di, the TEA substitution matrix for TEA, and corresponding substitution matrices built as described in Section A.3 for the different alphabet sizes. Combinations of alphabets use a matrix $\frac{1}{n}\sum_a^n M_a$ where $M_a$ is the score matrix constructed for alphabet $a$ and $n$ the number of alphabets combined. All alignments used gap open and extend penalties of 12 and 1.

## A.5    BENCHMARKS

The SCOPe40 benchmark is defined and evaluated as described in (Van Kempen et al., 2024), except applied to the latest version of SCOPe40 (v2.08) consisting of 15,128 single domains with a median length of 149 residues. The family and superfamily benchmark plots measure the sensitivity of detected TPs (same family, and same superfamily but not same family respectively) up to the first FP (where FP is defined as hit from a different fold). For 4-fold cross-validation results (labelled as CV in the figures), the sensitivity curves and AUCs are calculated only for comparisons between domains in the held-out folds. For all methods, each query was allowed to have up to 2000 hits (controlled by the -max-seqs parameter in MMseqs2 and Foldseek).

The reference-free multidomain benchmark is adapted from (Van Kempen et al., 2024) - we aligned all 688,852 community representatives from UniProt3D (Durairaj et al., 2023) using BLAST (2.5.0+) with an $E$ value threshold $< 10^{-3}$ and then clustered using SPICi (Jiang & Singh, 2010), resulting in 56,574 clusters. For each cluster, we picked the longest protein as representative. We randomly selected 100 representatives as queries and searched the set of remaining structures. Foldseek and MMseqs2 searches were run as described in (Van Kempen et al., 2024), while TEA search was run using the *exact* mode described in A.4. Evaluation was performed using the scripts obtained from github.com/steineggerlab/foldseek-analysis.

## A.6    E VALUES

We developed an empirical E-value model for the combined TEA and amino acid scoring system, where BLOSUM62 substitution scores are combined with a weight of 1.4 with TEA substitution scores. Standard Karlin-Altschul statistics assume a Gumbel-distributed score distribution, which does not hold for structural alphabet alignments due to convergent evolution of secondary and tertiary structure motifs producing a heavy-tailed false positive distribution (Edgar & Sahakyan, 2025). Instead, we adopted the proposed log-linear model where the cumulative false positive distribution $C(s|FP)$ is approximated as $log_{10}(C(s|FP)) = ms + c$, giving E-values of the form $E(s) = \frac{H}{Q}10^{(ms+c)}$, where $s$ is the coverage-weighted alignment score ($raw\sqrt{min(qcov, tcov)}$), H/Q is the average number of reported hits per query (computed at runtime from the prefilter output), and $m$ and $c$ are parameters fitted by log-linear regression of the empirical cumulative false positive distribution on the SCOP40c dataset, a curated subset of SCOPe v1.75 with 9,705 domains (Edgar & Sahakyan, 2025). False positives were defined as pairs from different SCOP folds; pairs within the same fold but different superfamilies were excluded. The log-linear parameters were fitted on the FPEPQ (false positive errors per query) range of 0.1–10, yielding $m = 0.017364$ and $c = 0.7636$ ($R = 0.99$). STEAM results are sorted by E-value (ascending), which corresponds to descending coverage-weighted score.

We evaluated the effect of E-value calibration on homology detection sensitivity using an all-against-all search on SCOP40c with -max-seqs set to 2000. (Table 1). Without E-value filtering, STEAM achieved sensitivities comparable to Foldseek and to TEA-only search using MMseqs2. When applying an E-value threshold of E < 10, STEAM retained high sensitivities at an empirical false positive rate of 11.2 FPEPQ, confirming the calibration accuracy of the log-linear model. To enable a fair comparison, we determined the Foldseek E-value threshold at which the empirical FPEPQ matched STEAM's rate of 11.2 and filtered Foldseek's results accordingly. At matched false positive

Table 1: Homology detection sensitivity on SCOP40c ($N = 2{,}875$ evaluable queries). FAM and SFAM report mean family and superfamily sensitivity before the first false positive (different fold). FPEPQ is the empirical false positive errors per query.

| Method | E-value filter | FAM | SFAM | FPEPQ |
|---|---|---|---|---|
| Foldseek | $E < 10$ (native) | 0.877 | 0.497 | 447.7 |
| STEAM (full) | $E < 10,000$ | 0.865 | 0.506 | 1702.4 |
| TEA (full) | none | 0.859 | 0.505 | 251.5 |
| STEAM | $E < 10$ | 0.825 | 0.422 | 11.2 |
| Foldseek (corrected) | matched FPEPQ | 0.820 | 0.402 | 11.2 |

rates, STEAM's calibrated E-values provide comparable or slightly better sensitivity to Foldseek while offering meaningful statistical significance estimates. Notably, Foldseek's native E-values substantially overestimate significance on this dataset, with an FPEPQ of 447.7 at its reported E < 10 threshold, consistent with previous observations that structural alphabet E-values based on Gumbel distributions fail to account for the heavy-tailed false positive score distribution arising from convergent structural evolution (Edgar & Sahakyan, 2025).

## A.7 WEB SERVER

A website was developed to facilitate access to TEA-converted datasets, currently offering downloads in FASTA format for UniRef50 (version 2025_03) (Suzek et al., 2007) and AlphaFold Clusters (afdbv4) (Barrio-Hernandez et al., 2023), with additional sources planned for future inclusion. Users may also submit amino acid sequences directly for conversion into TEA sequences (on CPU), which are queried against the available datasets using an in-house implementation of STEAM. Results are queried via FastAPI (Ramírez), with results persisted in MongoDB (Chodorow, 2013). The interface is built with Next.js (Vercel, 2016), and where a search hit has a known 3D structure, it is fetched from the MongoDB and visualised interactively using the NGL structure viewer (Rose & Hildebrand, 2015).

## A.8 40K-PLDDT SET

Sequences and 3Di characters for the character comparison analysis were obtained from the SCOPe40 domains, and from 40,000 proteins with lengths between 100 and 600 selected randomly from the Foldseek `alphafold_uniprot50` database such that 10,000 proteins each had an average AFDB pLDDT within the bins 0-50 ("Very low"), 50-70 ("Low"), 70-90 ("High") and 90-100 ("Very high"). ESMFold (Lin et al., 2023) structures were created for all proteins in this set.

## A.9 AFDB CLUSTERS

We obtained the UniProt accessions corresponding to AFDB cluster representatives from the `5-allmembers-repId-entryId-cluFlag-taxId.tsv` file in afdb-cluster.steineggerlab.workers.dev and restricted representatives to those with `cluFlag` set to 2 (singletons) and 4 (non-singleton representatives). These sequences were converted to TEA sequences and filtered with an average TEA entropy threshold of 0.25. The search was run for the singletons passing the threshold against all sequences passing the threshold with a coverage threshold of 90% across both query and target, and a minimum sequence identity threshold of 50%.

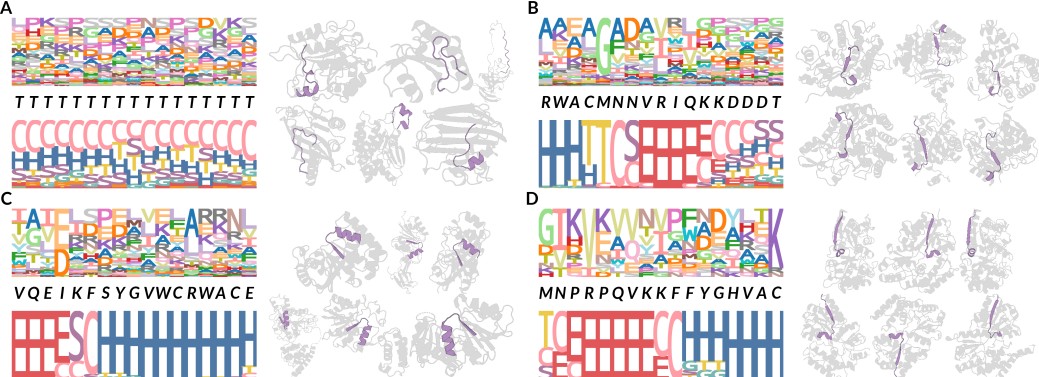

Figure 5: **Conserved TEA motifs.** Examples of TEA motifs of length 17 found **A)** in 76 SCOPe40 IDs across 56 different folds, **B)** in 261 IDs across 26 superfamilies only in fold c.1 **C)** in 79 IDs across 40 families only in superfamily c.66.1, and **D)** 17 IDs only in family c.94.1.0 are shown. For each motif, the left panel shows the amino acid frequency logo, the TEA characters of the motif in italics, and the DSSP frequency logos for all occurrences in SCOPe40, and the right panel shows examples of structures (gray) containing the motif (highlighted in purple). DSSP labels are as follows, H: $\alpha$-helix, G: 3-10 helix, E: $\beta$-sheet, C: random coil, T: H-bonded turn, S: bend.

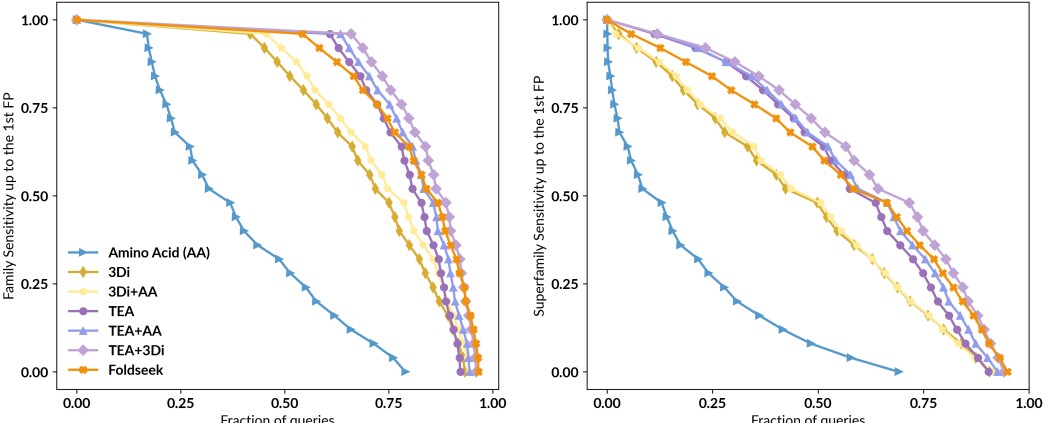

Figure 6: **Exhaustive alignments of alphabet combinations.** **A)** Cumulative distributions of sensitivity for homology detection on the SCOPe40 database of single-domain structures. TPs are matches within the same family; FPs are matches between different folds. Sensitivity is the area under the ROC curve up to the first FP. **B)** Same as A with TPs as matches within the same superfamily.

|  | A | C | D | E | F | G | H | I | K | L | M | N | P | Q | R | S | T | V | W | Y |
|---|---|---|---|---|---|---|---|---|---|---|---|---|---|---|---|---|---|---|---|---|
| A | 7.4 | 1.1 | -6.3 | -5.1 | -6.3 | -2.7 | -1.8 | -8.5 | -11.6 | -6.3 | -2.5 | -8.1 | -8.4 | -8.9 | -6.3 | -4.0 | -2.3 | -4.5 | 1.0 | -2.7 |
| C | 1.1 | 7.7 | -8.3 | -2.1 | -7.1 | -3.1 | 0.1 | -7.8 | -12.3 | -7.8 | -2.0 | -8.2 | -9.6 | -9.3 | -2.0 | -6.1 | -3.4 | -5.9 | -3.2 | -6.1 |
| D | -6.3 | -8.3 | 9.3 | -6.3 | -3.3 | -4.3 | -3.0 | -3.5 | -1.0 | -1.3 | -5.6 | -6.6 | -7.6 | -5.5 | -6.5 | -3.0 | -0.6 | -7.1 | -6.8 | -4.0 |
| E | -5.1 | -2.1 | -6.3 | 8.1 | -4.7 | -8.9 | -3.1 | -5.9 | -5.1 | -8.2 | -3.6 | -7.1 | -5.1 | -3.7 | -3.7 | -6.8 | -5.2 | -3.8 | -2.2 | -8.4 |
| F | -6.3 | -7.1 | -3.3 | -4.7 | 8.1 | -1.9 | -1.0 | -2.3 | -3.1 | -4.5 | -4.9 | -6.5 | -3.6 | -0.9 | -7.7 | 0.7 | 0.0 | -5.3 | -5.3 | -1.2 |
| G | -2.7 | -3.1 | -4.3 | -8.9 | -1.9 | 8.0 | -0.2 | -5.3 | -6.0 | -4.3 | -2.2 | -3.4 | -3.3 | -4.2 | -5.1 | -2.0 | -0.8 | -4.2 | -5.6 | -0.2 |
| H | -1.8 | 0.1 | -3.0 | -3.1 | -1.0 | -0.2 | 8.3 | -3.7 | -6.7 | -3.2 | -3.5 | -4.8 | -4.5 | -3.5 | -4.0 | 0.5 | 1.8 | -0.5 | -0.7 | -2.1 |
| I | -8.5 | -7.8 | -3.5 | -5.9 | -2.3 | -5.3 | -3.7 | 9.1 | -3.2 | -3.1 | -5.4 | -8.0 | -5.8 | -2.5 | -7.6 | -3.4 | -2.5 | -7.1 | -8.3 | -0.6 |
| K | -11.6 | -12.3 | -1.0 | -5.1 | -3.1 | -6.0 | -6.7 | -3.2 | 8.9 | -3.8 | -7.7 | -8.0 | -5.6 | -4.1 | -11.9 | -5.1 | -3.5 | -6.8 | -11.8 | -6.9 |
| L | -6.3 | -7.8 | -1.3 | -8.2 | -4.5 | -4.3 | -3.2 | -3.1 | -3.8 | 9.7 | -3.6 | -5.5 | -5.6 | -6.4 | -9.6 | -4.4 | -1.3 | -7.0 | -6.6 | -4.6 |
| M | -2.5 | -2.0 | -5.6 | -3.6 | -4.9 | -2.2 | -3.5 | -5.4 | -7.7 | -3.6 | 7.8 | -1.2 | -4.2 | -6.9 | -9.7 | -4.0 | -0.4 | -9.1 | -3.4 | -2.2 |
| N | -8.1 | -8.2 | -6.6 | -7.1 | -6.5 | -3.4 | -4.8 | -8.0 | -8.0 | -5.5 | -1.2 | 8.0 | -1.0 | -7.2 | -5.4 | -6.2 | -5.5 | -4.3 | -7.4 | -6.8 |
| P | -8.4 | -9.6 | -7.6 | -5.1 | -3.6 | -3.3 | -4.5 | -5.8 | -5.6 | -5.6 | -4.2 | -1.0 | 8.0 | -3.7 | -3.2 | -6.2 | -5.0 | -3.7 | -9.7 | -7.6 |
| Q | -8.9 | -9.3 | -5.5 | -3.7 | -0.9 | -4.2 | -3.5 | -2.5 | -4.1 | -6.4 | -6.9 | -7.2 | -3.7 | 8.1 | -5.8 | -3.5 | -1.9 | -3.3 | -9.1 | -4.0 |
| R | -6.3 | -2.0 | -6.5 | -3.7 | -7.7 | -5.1 | -4.0 | -7.6 | -11.9 | -9.6 | -9.7 | -5.4 | -3.2 | -5.8 | 7.9 | -7.7 | -11.9 | -1.5 | -5.2 | -10.7 |
| S | -4.0 | -6.1 | -3.0 | -6.8 | 0.7 | -2.0 | 0.5 | -3.4 | -5.1 | -4.4 | -4.0 | -6.2 | -6.2 | -3.5 | -7.7 | 8.7 | 0.3 | -3.2 | -2.4 | 1.0 |
| T | -2.3 | -3.4 | -0.6 | -5.2 | 0.0 | -0.8 | 1.8 | -2.5 | -3.5 | -1.3 | -0.4 | -5.5 | -5.0 | -1.9 | -11.9 | 0.3 | 8.0 | -6.2 | -2.6 | 1.2 |
| V | -4.5 | -5.9 | -7.1 | -3.8 | -5.3 | -4.2 | -0.5 | -7.1 | -6.8 | -7.0 | -9.1 | -4.3 | -3.7 | -3.3 | -1.5 | -3.2 | -6.2 | 7.9 | -0.6 | -5.8 |
| W | 1.0 | -3.2 | -6.8 | -2.2 | -5.3 | -5.6 | -0.7 | -8.3 | -11.8 | -6.6 | -3.4 | -7.4 | -9.7 | -9.1 | -5.2 | -2.4 | -2.6 | -0.6 | 7.4 | -2.5 |
| Y | -2.7 | -6.1 | -4.0 | -8.4 | -1.2 | -0.2 | -2.1 | -0.6 | -6.9 | -4.6 | -2.2 | -6.8 | -7.6 | -4.0 | -10.7 | 1.0 | 1.2 | -5.8 | -2.5 | 8.2 |

Figure 7: TEA **substitution matrix.** See Section A.3 for how the matrix is computed. The numbers are colored on a **red**-**blue** color scale with **red** being the lowest and **blue** the highest.

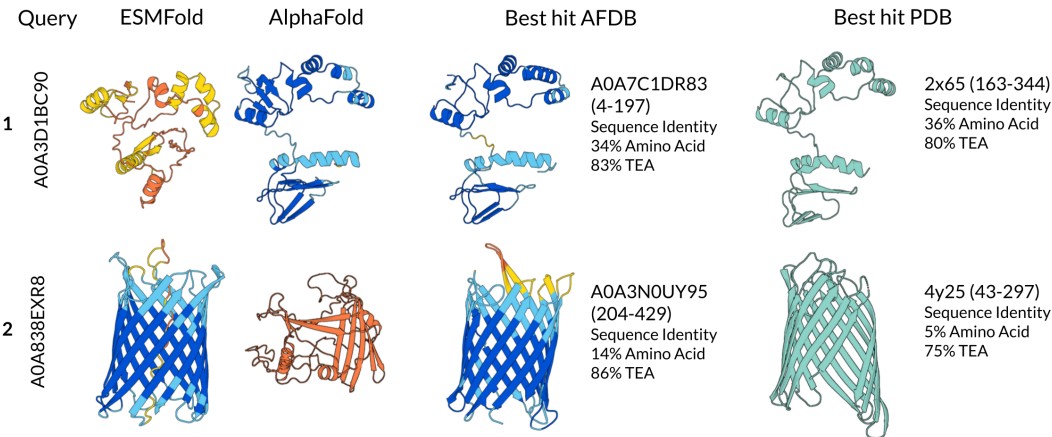

Figure 8: **TEA confidently represents the correct structure even in AlphaFold and ESMFold failure modes** Two examples of searches for proteins highlighted in Figure 3C-D, displayed as the ESMFold and AlphaFold structures of the protein respectively, and then the closest TEA hit for the protein in AFDB Clusters and in the PDB respectively, both cropped to the alignment region with residue range displayed. The amino acid and TEA sequence identities of the TEA alignments are also displayed.

| Method | `max-seqs` | Time (s) | Family AUC | Superfamily AUC | Fold AUC |
|---|---|---|---|---|---|
| MMseqs2 | 300 | 8 | 0.192 | 0.07 | 0 |
| | 1000 | 5 | 0.192 | 0.07 | 0 |
| TEA (exact) | 300 | 26 | 0.789 | 0.561 | 0.072 |
| | 1000 | 80 | 0.806 | 0.585 | 0.102 |
| TEA (sensitive) | 300 | 33 | 0.787 | 0.560 | 0.075 |
| | 1000 | 89 | 0.808 | 0.588 | 0.103 |
| Foldseek | 300 | 69 | 0.764 | 0.512 | 0.075 |
| | 1000 | 87 | 0.805 | 0.559 | 0.095 |
| EBA | - | 7,560 | 0.84 | 0.6 | 0.079 |

Table 2: **Runtime and performance for SCOPe40 (15,128 proteins).** The `max-seqs` column describes the number of hits allowed to pass the pre-filtering step. EBA does not have a pre-filtering step and thus all pairs of alignments are computed and then sorted to take the top 1000 per query. All methods were run with 64G RAM and 128 cores.