# OpenReview forum: "Rewriting protein alphabets with language models"
_ICLR.cc/2026/Workshop/LMRL — ICLR 2026 Workshop LMRL Poster_

### Official Review · Reviewer_oPd4 · 2026-02-21
**homology detection using discrete reduction of protein language model embeddings**

**Rating:** 7
**Confidence:** 4

**Review:**

Overall this paper describes an interesting contrastive learning approach to reduce protein language model embeddings to 20 discrete characters that can then be used for rapid homology searches (using string-based matching).

The authors approach seems inspired by foldseek, which has been a major breakthrough in structural similarity search. This paper addresses an important topic in computational biology and is clearly written and presented, including comparisons to standard tools such as foldseek and mmseqs2.

There are at least two major issues with the paper that need to be addressed.

1. While the contrastive learning approach here is new (to my knowledge), the idea to discretize protein language model embeddings to speed up remote homology detection is not new. To my knoweldge it was first published in 2023:  Kilinc et al. Improved global protein homolog detection with major gains in function identification PNAS 2023 .The authors don't discuss or compare with this previous work that used a different approach to discretize protein language model embeddings (discrete cosine transforms) for remote homology detection.

2. The authors claim that their method can be done with little memory on limited hardware. But it is not clear if they are factoring in the compute resources needed to get the ESM2 embeddings in the first place. To make a 'fair' comparison with sequence-based approaches, this must be considered.

---

### Official Review · Reviewer_QLcM · 2026-02-23
**Review for “Rewriting protein alphabets with language models”**

**Rating:** 9
**Confidence:** 4

**Review:**

Summary:

This paper introduces TEA, a learned 20-letter structure-based discretization of protein language model (pLM) embeddings. This was done with a shallow projection head on structurally aligned residue triplets from SCOPe40 to map per-residue embeddings into discrete characters. Sequences labeled with TEA alphabet can then be aligned, thus enabling remote homology detection performance comparable to structure-based approaches like Foldseek, while retaining sequence-level efficiency and without requiring structural input. The authors further demonstrate that entropy of the predicted character distribution correlates with structural confidence (pLDDT), and they apply TEA at scale to improve AFDB clustering.

===================================

Strengths:

1.	TEA provides a simple yet effective framework for converting pLM embeddings into a structurally informed alphabet without explicitly requiring structural input at inference time. The approach is computationally lightweight, is adaptable across several different pLMs, and integrates with existing sequence alignment infrastructure.

2.	The authors perform careful cross-validation at the fold level and provide comparisons against MMseqs2, Foldseek, and EBA.

3.	TEA achieves performance comparable to Foldseek on SCOPe40 and multi-domain benchmarks. The combination of TEA and 3Di further improves performance, suggesting complementarity rather than redundancy.

4.	Entropy is a useful heuristic that correlates with pLDDT and search sensitivity to provide a useful method for filtering unreliable searches or alignments.

5.	The application to AFDB clustering demonstrates scalability and potential real practicality. The reported annotation agreement among new protein clusters is very encouraging.

===================================

Weaknesses:

1.	The training objective explicitly includes entropy minimization and uniform usage regularization. It is unclear how sensitive results are to the weighting of these terms.

a.	Does adjusting these weights lead to mode collapse (such as overuse of a small subset of characters)?

b.	Does removing or reweighting the entropy loss alter the entropy-pLDDT correlation observed in Section 2.3?

An ablation would strengthen confidence in the robustness of the entropy-based claims.

2.	While entropy correlates strongly with pLDDT, it reflects uncertainty in the discretization head rather than structural disorder per se. High entropy could arise from intrinsic disorder, weak evolutionary constraint, embedding degeneracy, or insufficient representation power. It would strengthen the analysis to disentangle these effects, for example by evaluating entropy on experimentally validated disordered regions or conserved but flexible motifs. Please feel free to clarify if you disagree with this perspective.

3.	TEA performs particularly well in cases where structure prediction is unreliable. A more specific comparison between TEA and Foldseek on low-pLDDT proteins would help quantify this advantage and better support this claim.

4.	Figure 2E + F would benefit from clearer explanation or illustration of evaluation procedure.

5.	The AFDB analysis relies on TEA identity (>50%) thresholds. It may be informative to show how cluster recovery and annotation agreement vary as a function of TEA identity threshold to better understand robustness.

6.	This might be outside the scope of the paper and will defer to the discretion of the authors, but may be interesting to see if TEA is sensitive to fitness (zero-shot analysis) by benchmarking on ProteinGym - compute the WT TEA tokens and calculate the log-likelihood ratios with mutant TEA tokens. Although I do not intuitively know how different the WT TEA tokens will be compared to mutants TEA tokens.

7.	The paper benchmarks primarily against alignment-based approaches. It may strengthen the evaluation to compare against scalable multimodal representation methods such as ProTrek [1], BioCLIP [2], BioM3 [3], or related embedding retrieval systems that also demonstrate remote homology detection capability at scale. This would clarify TEA’s positioning relative to modern embedding search methods.

[1] Su, J., He, Y., You, S., Jiang, S., Zhou, X., Zhang, X., ... & Yuan, F. (2025). A trimodal protein language model enables advanced protein searches. Nature Biotechnology, 1-7.
[2] Robinson, L., Atkinson, T., Copoiu, L., Bordes, P., Pierrot, T., & Barrett, T. D. (2023). Contrasting sequence with structure: Pre-training graph representations with PLMs. bioRxiv, 2023-12.
[3] Praljak, N., Yeh, H., Moore, M., Socolich, M., Ranganathan, R., & Ferguson, A. L. (2024). Natural language prompts guide the design of novel functional protein sequences. bioRxiv, 2024-11.

===================================

Overall:

Simple and effective method that uses discretized pLM embeddings for structure-aware homology search within sequence alignment frameworks. Empirical results are strong and demonstrate competitiveness with structure-based methods while maintaining scalability. It would be valuable to see deeper analysis of entropy robustness and calibration, more systematic validation on poorly predicted structures, and comparisons to modern embedding-based retrieval approaches. Overall, a practically impactful contribution with clear potential for broader applications in large-scale protein annotation, clustering, and opportunities for structure guided design.

---

### Official Review · Reviewer_BMUu · 2026-02-25
**This paper tried to bridges the gap between sequence and structure through discretizing protein language model embeddings**

**Rating:** 7
**Confidence:** 3

**Review:**

This work presents a technically sound and well-validated method for discretizing protein language model embeddings into a compact 20-letter alphabet that enables fast and sensitive remote homology detection using standard sequence tools such as MMseqs2, while achieving performance comparable to structure-based approaches like Foldseek. The paper includes thorough benchmarking, cross-validation on unseen folds, and large-scale application to AlphaFold DB.The main contribution is the integration of contrastive learning with embedding discretization to bridge modern protein language models and classical alignment frameworks, offering both conceptual insight and practical utility. The significance is for scalable homology search and functional annotation without structural input. Key strengths include strong empirical performance, compatibility with existing infrastructure, built-in entropy-based confidence estimation, and demonstrated complementarity with structural representations. Limitations include reliance on structurally supervised training data, limited theoretical justification for the fixed 20-letter alphabet size, potential sensitivity to substitution matrix design, and a lack of direct experimental validation of some proposed downstream applications like improved MSAs or structure prediction. Overall, the work represents a meaningful and practically impactful advance in computational protein analysis. However, this paper is not relavant to LMRL theme and I would like to make a note here.

---

### Meta-Review · Area_Chair_qwdC · 2026-02-27

**Recommendation:** Accept (Poster)
**Confidence:** 4

**Metareview:**

Consistently strong reviews and well aligned with the workshop.

---

### Decision · Program_Chairs · 2026-03-02

**Decision:**

Accept (Poster)

**Comment:**

Please see the meta-review.